# Anchoring Fine-tuning of Sentence Transformer with Semantic Label Information for Efficient Truly Few-shot Classification

**Amalie Brogaard Pauli**[1]     **Leon Derczynski**[2]     **Ira Assent**[1]

[1]Department of Computer Science, Aarhus University, Denmark
[2]IT University of Copenhagen, Denmark
{ampa,ira}@cs.au.dk, ld@itu.dk

## Abstract

Few-shot classification is a powerful technique, but training requires substantial computing power and data. We propose an efficient method with small model sizes and less training data with only 2-8 training instances per class. Our proposed method, AncSetFit, targets low-data scenarios by anchoring the task and label information through sentence embeddings in fine-tuning a Sentence Transformer model. It uses contrastive learning and a triplet loss to enforce training instances of a class to be closest to its own textual semantic label information in the embedding space - and thereby learning to embed different class instances more distinct. AncSetFit obtains strong performance in data-sparse scenarios compared to existing methods across SST-5, Emotion detection, and AG News data, even with just two examples per class.

## 1 Introduction

Data sparsity remains a problem for training NLP models. Pre-trained language models (PLMs) have been adapted to NLP downstream tasks using only a few ground truth examples. However, many of these methods require substantial computing, reducing their practicality. The SetFit method (Tunstall et al., 2022) offers a promising route to few-shot text classification, due to its computational efficiency and relatively low data requirement. The method is prompt-free, which is an advantage in that it skips the variation inherent to prompt-based learning methods since these depend on choosing good/multiple prompts. On the other hand, in a truly few-shot setting, providing the model with information about the labels it should generalise to from the few given training examples is expected to be helpful (Schick and Schütze, 2020).

We ask (RQ1) if the efficiency of small models in SetFit can be retained while also making the method label-aware and if this improves performance, especially in cases with really little training data. To do this, we propose adding semantic information about the task and labels using fine-tuning of sentence transformers (Reimers and Gurevych, 2019). Through this, we *inform* the model of what is to be classified by adding textual information. We use contrastive learning and a triplet loss to enforce instances from different classes to be closest to their own class' textual description. Additionally, (RQ2) we ask to what extent variation in textual information affects model performance. Our contributions include:

- AncSetFit, a method that adds semantic information about the task to SetFit while maintaining the same efficiency.

- Empirical evidence that AncSetFit gives higher performance in extremely few-shot classification settings.

Code is publicly available at https://github.com/AmaliePauli/AncSetfit.

## 2 Background

### 2.1 Related work

Few-shot text classification reduces the need for annotated data, which is often costly and time-consuming to obtain. It has gained increased attention with large pre-trained language models (PLMs), especially GPT-3 with in-context learning, which has no parameter update to the model, but a model size of 175 billion parameters (Brown et al., 2020). Prompting is also used in Pattern Exploiting Training (PET) (Schick and Schütze, 2020), which uses the training objective of the language model with cloze-like questions to predict a task. ADAPET extends PET for improved performance and eliminating the need for unlabeled task-specific data (Tam et al., 2021), based on AL-BERT (Lan et al.). PERFECT is likewise a PET-based method, which improves efficiency using task-specific adapters and eliminates the need for

handcrafted verbalizers and patterns (Mahabadi et al., 2022).

Recently, there has been work to reduce runtime and model sizes. The state-of-the-art in few-shot classification is T-Few (Liu et al., 2022) which is a parameter-efficient fine-tuning (PEFT) method based on a T0 model (Sanh et al., 2021). Although T-Few is much smaller than GPT-3, it is still large with billions of parameters. A method that is truly efficient in terms of computing requirements is the SetFit method, which uses a backbone model only in the sizes of millions of parameters (Tunstall et al., 2022). In our work, we build on SetFit to address challenging few-shot classification tasks efficiently.

## 2.2 SetFit: An efficient method

SetFit (Tunstall et al., 2022) uses a pre-trained Sentence Transformer (Reimers and Gurevych, 2019). The sentence transformer embeds entire sentences-/paragraphs in one embedding space based on similarity measures and siamese and triplet networks. SetFit has two phases: **1)** it applies contrastive learning to fine-tune a sentence transformer model. It augments training data by generating triplets from the few available training samples: drawing pairs of samples from the same class and pairs from distinct classes and minimizing the loss function $||\mathbb{1}(u,v) - d(e(u), e(v))||_2$ where $d(\cdot, \cdot)$ is the cosine distance function, $e(\cdot)$ is the embedding, $u, v$ are input samples and the indicator function $\mathbb{1}(u, v) := \{1 \text{ if } L(u) = L(v) \text{ else } 0$, where $L$ is a mapping function to a class; so 1 indicates the two samples are of the same class (positive) and 0 the samples are of different classes (negative). Intuitively, this clusters embeddings such that embeddings of different classes should be far apart, and those of the same class close in the embedding space. **2)** SetFit uses a traditional classifier on the embedding vectors.

## 3 AncSetFit: Anchoring SetFit with Semantic Label Information

The intuition in SetFit, as discussed in the section above, is to get the embeddings to separate class-wise before inputting them into a classifier. However, if the training data is really small, there might be a lack of information about the classes that would allow the embedding to cluster accordingly and thus to generalise. When only a few samples are available, the samples might have other properties in common than what the task prescribes.

We see the benefit of adding task description information, as seen in PET methods. We, therefore, propose to inject information about the task into the method, such that it can help separate the embeddings in the desired directions for a task. Intuitively, we want to provide the model with good class-specific starting points in the embedding space for sentences to cluster around. We think of them as starting points since we want the class-specific points to be learnable as well.

We introduce *anchor statements*, which are sentences containing semantic information for each class i.e. a task/class description. As they are textual, they can be embedded with the model and hence updated under fine-tuning. We look to the Sentence Transformer paper (Reimers and Gurevych, 2019), which experiments with a triplet loss function for sentence similarity in articles. The input is a triplet of three sentences, two from the same article, and one from a distinct, and then the loss minimises the distance between the same class pair. We propose to modify the input of the loss function to be a class-specific fixed anchor statement, and two samples from distinct classes, one matching the class of the anchor statement - thus getting the semantic label information into the loss function. The objective is to ensure that the samples of one class are closer to their own fixed class-specific anchor statement, hence getting the embeddings to separate in the desired direction.

More formally, we generate input triplets $(a, u, v)$ by drawing samples $u, v$ from two distinct classes and adding the anchor statement $a$ of one of the classes. We use the loss function:

$$\max(d(e(a^p), e(u^p)) - d(e(a^p), e(v^n)) + \epsilon, 0)$$

where $p$ is indicating samples of the same class (positive), and $n$ of another class (negative), $d(\cdot, \cdot)$ is the cosine distance function, $e(\cdot)$ is the embedding, and the margin $\epsilon$ is a hyperparameter. Essentially, we want to achieve $d(e(a^p), e(u^p)) + \epsilon < d(e(a^p), e(v^n))$, meaning that the sentences from a class should be closer to the anchor statement i.e the class description of that class than a sentence from a different class should be. Therefore, the $\epsilon$ is the margin to the anchor point we want to enforce between samples of different classes. Note that during training, the embedding of the anchor statement is also updated, and the pre-set 'centre points' can therefore adjust accordingly. For intuition, see also a visualisation on fine-tuning two datasets using PCA in Appendix A.

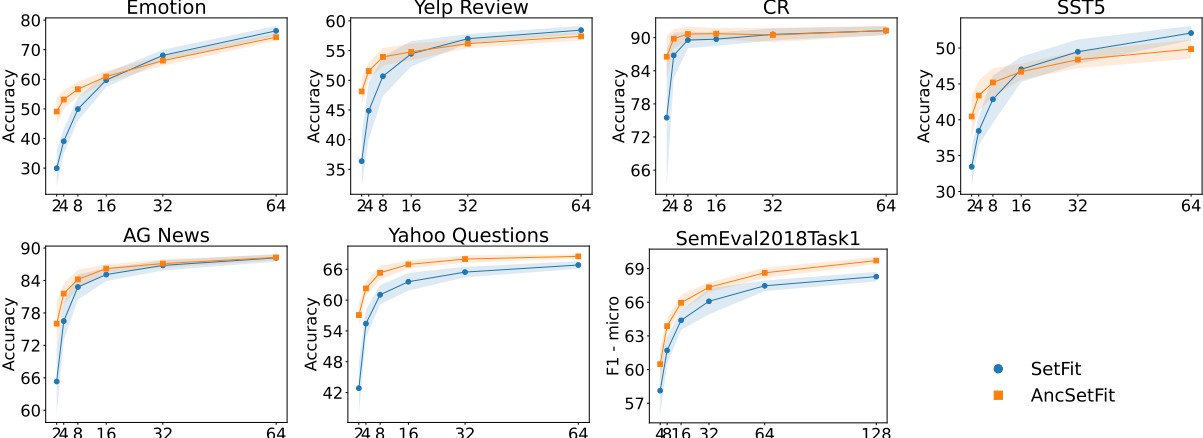

Figure 1: Average accuracy for the single-label tasks and F1-micro for the multi-label task, and ± std deviation for different training set sizes for comparing SetFit and AncSetFit. Training data sizes are per class.

## 4 Experiments

### 4.1 Data

We conduct experiments on text classification data, where several have been used in mimic few-shot settings (Min et al., 2022; Gao et al., 2021; Tunstall et al., 2022). In a true few-shot setting, data for developing and tuning hyperparameters would not be available. Thus, as in Tunstall et al. (2022), we split the data sets into development and testing. All data are in English, mainly in sentiment analysis and topic classification; mostly single-label classification, plus a few multi-label tasks.

- **Data for testing:** Customer Review (CR) (Hu and Liu, 2004); Stanford Sentiment Treebank-5 (SST5) (Socher et al., 2013); Emotion (Saravia et al., 2018); AG News, Yahoo Answer Topics (Yahoo Questions) and Yelp Review (Zhang et al., 2015); multi-label SemEval20028Task1 (Mohammad et al., 2018).

For further details on data and tasks see Appendix B. In total, we use seven different datasets for development and hyperparameter tuning - but stay as similar to SetFit as possible for comparison. We experiment with different hyperparameter settings, such as the margin parameter and anchor statements on development data. We report experiments examining the effect of different anchor statements on the following development dataset SST2 (Socher et al., 2013), IMDB (Maas et al., 2011), Student Question Categories,[1] BBC News (Greene and Cunningham, 2006), see Section 6. After the development stage, we have one go at the test datasets.

We simulate the few-shot setting by drawing samples with balanced labels, as in Gao et al. (2021); Tunstall et al. (2022) randomly from the original training data, repeated 10 or 20 times for each experiment, to account for possible variance in data quality. Finally, we test our method on the real-world few-shot classification benchmark RAFT (Alex et al., 2021) [2] with 11 tasks, each of 50 training samples.

### 4.2 Testing Setup and baselines

Mainly, we benchmark against the SetFit method since AncSetFit is a proposed alternative, and study the effect of our proposed anchor statements in isolation. Note that both methods use the same relatively small backbone model and are equal in terms of efficiency. Appendix D gives training details and hyperparameters, Appendix C anchor statements.

| Method - backbone | Size |
|---|---|
| **PERFECT** roberta-large | 355M |
| **ADAPET_XXL** albert-xxlarge-v2 | 223M |
| **ADAPET_BASE** albert-base-v2 | 117M |
| **SetFit** paraphrase-mpnet-base-v2 | 110M |
| **AncSetfit** paraphrase-mpnet-base-v2 | 110M |

Table 1: Backbone model size in terms of number of parameters. Backbone models are downloaded from HuggingFace, URLs in Appendix D

[1]kaggle.com/datasets/mrutyunjaybiswal/iitjee-neet-aims-students-questions-data

[2]huggingface.co/spaces/ought/raft-leaderboard

| | Size | PERFECT | ADAPET_BASE | ADAPET_XXLARGE | SetFit | AncSetFit |
|---|---|---|---|---|---|---|
| **Emotion** | 2 | 22.1 ±6.0 | 34.5 ±6.7 | 46.7 ±7.0 | 30.0 ±6.2 | **49.1** ±4.7 |
| | 4 | 22.5 ±3.5 | 45.0 ±5.6 | 48.6 ±4.9 | 39.1 ±4.2 | **53.2** ±3.2 |
| | 8 | 30.2 ±6.8 | 50.9 ±3.6 | 53.9 ±3.9 | 49.9 ±4.0 | **56.6** ±2.6 |
| **SST5** | 2 | 27.3 ±3.1 | 32.9 ±3.2 | **47.0** ±3.2 | 33.4 ±2.8 | 40.5 ±3.0 |
| | 4 | 30.3 ±5.5 | 35.5 ±2.0 | **48.1** ±3.2 | 38.4 ±2.0 | 43.4 ±2.1 |
| | 8 | 34.8 ±2.7 | 39.6 ±2.0 | **49.7** ±2.0 | 42.8 ±3.4 | 45.2 ±2.0 |
| **CR** | 2 | 63.7 ±8.6 | 76.8 ±10.1 | **87.2** ±5.3 | 75.5 ±12.6 | 86.6 ±3.9 |
| | 4 | 75.6 ±7.7 | 77.0 ±8.3 | **90.6** ±2.5 | 86.8 ±4.2 | 89.8 ±1.8 |
| | 8 | 82.7 ±9.2 | 78.6 ±6.1 | 90.6 ±1.7 | 89.6 ±1.5 | **90.7** ±1.4 |
| **AG News** | 2 | 51.8 ±12.5 | 64.5 ±2.6 | **76.8** ±6.5 | 65.3 ±6.2 | 76.0 ±4.2 |
| | 4 | 67.2 ±8.3 | 73.1 ±3.2 | **84.1** ±2.0 | 76.5 ±3.6 | 81.6 ±1.9 |
| | 8 | 79.2 ±5.6 | 77.9 ±3.3 | **86.0** ±2.8 | 82.8 ±2.4 | 84.2 ±1.8 |
| **Yahoo Q\*** | 2 | 28.1 ±4.9 | 48.8 ±3.1 | 51.1 ±3.2 | 42.9 ±5.2 | **57.1** ±2.3 |
| | 4 | 40.3 ±4.7 | 56.6 ±2.1 | 59.0 ±1.8 | 55.4 ±2.7 | **62.3** ±1.4 |
| | 8 | 54.2 ±2.9 | 59.0 ±1.8 | 63.5 ±1.9 | 61.0 ±1.9 | **65.3** ±1.4 |
| **Yelp R\*** | 2 | 24.7 ±3.8 | 32.0 ±4.6 | **50.9** ±1.2 | 36.4 ±4.3 | 48.1 ±2.3 |
| | 4 | 30.5 ±5.0 | 40.8 ±3.3 | **54.7** ±2.2 | 44.9 ±4.8 | 51.6 ±1.4 |
| | 8 | 39.1 ±3.5 | 44.0 ±3.1 | **58.2** ±1.6 | 50.7 ±3.4 | 53.9 ±1.5 |

Table 2: Average accuracy, standard deviation for 2,4 and 8 training samples per class. *Yahoo Q: Yahoo Questions, Yelp R: Yelp Review

AncSetFit is targeting small model sizes and extreme few-shot settings, we therefore also provide benchmark results on ADAPET (Tam et al., 2021) and PERFECT (Mahabadi et al., 2022) when training on 2-8 instances per class. ADAPET and PERFECT are chosen as benchmarks as these methods also use relatively small back-bone model sizes in terms of the number of parameters, see Table 1. ADAPET originally used ALBERT xxlarge-v2 as the backbone, but in order to compare the methods with similar model sizes, we also benchmark again ADAPET using ALBERT base-v2. With ADAPET we maintain the same level of information provided to the model as in AncSetFit, with the information from the anchor statements corresponding to the verbalisers and patterns. Note PERFECT and ADAPET originally use more data, PERFECT is developed using 32 training instances per class. ADAPET originally assumed access to a development set, which is disabled here, similar to the comparison done in Tunstall et al. (2022). The experiments are repeated 20 times for AncSetFit and SetFit, 10 times for ADAPET and PERFECT. Implementation and training details are provided in Appendix D.

## 5 Results and Discussion

We compare AncSetFit to SetFit on seven different datasets in Fig. 1, with selected details in Table 2. On all seven datasets, we see that AncSetfit outperforms SetFit in terms of accuracy when there are only 2-8 samples per class available for training, thus demonstrating its benefit for the most extreme cases. On larger sample sizes, the gain decreases. This result is consistent with the Ablation study of the anchor statements in Section 6: fine-tuning of the sentence transformer benefits from anchoring information when the training data size is small, and for larger training sizes the method becomes less and last no longer dependent on it. We notice large numeric differences in accuracy in the extremely few-shot cases; e.g., after training on only two samples times six classes in the Emotion dataset, AncSetFit achieves an average accuracy of 49.1 against the lower 30.0 for SetFit. We also observe a lower standard deviation across datasets for AncSetfit than for SetFit, which means that it is less dependent on getting good training samples.

As AncSetFit only holds an advantage in performances over SetFit when training data sizes are low, we compare AncSetFit to ADAPET and PER-

FECT in the training size range of 2-8 instances per class. (Comparing SetFit to ADAPET and PER-FECT for larger training sizes we refer the reader to (Tunstall et al., 2022).) AncSetFit outperforms PERFECT and ADAPET_BASE on all six datasets for training sizes 2,4,8 in Table 2. Comparing An-cSetfit to ADAPET_XXL, we get more varying results. AncSetFit is superior on the Emotion detection task and the topic classification in Yahoo Questions, whereas ADAPET_XXL is on SST5, AG News and Yelp Review.

However, ADAPET_XXL model size is double that of AncSetfit in terms of parameters Table 1, and AncSetFit completes both training and inference in seconds, whereas ADAPET_XXL takes hours in our setting. Numbers listed in Appendix E. Thus, AncSetFit is both efficient and effective.

**RAFT leaderboard** AnSetFit achieves higher performance in 4 out of 11 RAFT tasks than Set-Fit. SetFit shows a higher average accuracy. The 'Banking_77' dataset has classes with no training samples, which makes it unsuitable for supervised learning with AncSetFit. Further, the majority of the datasets are binary with more labels per class than in the range where our experiments show an acceleration in results. Lastly, we note that several tasks are semantically complex, in that the description of the classes is rather long, making it more difficult to capture the information in the anchor statement. For example, the task on 'systematic review inclusion' has a specification list of criteria for the classes. However, AncSetFit gets higher performance on the two Twitter tasks of detecting hate speech and complaints.

# 6 Ablation Study: Effect of the Anchor Statements

In the development phase, we study the impact of the anchor statements. We construct the anchor statement in the generic form where a template is specific to each task and $w$ is a word describing the class. We evaluate four ways of constructing $w$; 1) words of the labels, 2) a single letter representing no actual information, 3) permutated label words from 1 such that the label word does not match the class, and 4) label synonyms (see Appendix B Table 4 for templates). Figure 2 shows the results for 4,8,16,32 and 64 training samples per class, averaged over 10 runs. In all four tasks we observe for small training set sizes: 1) providing the model with the correct semantic label information

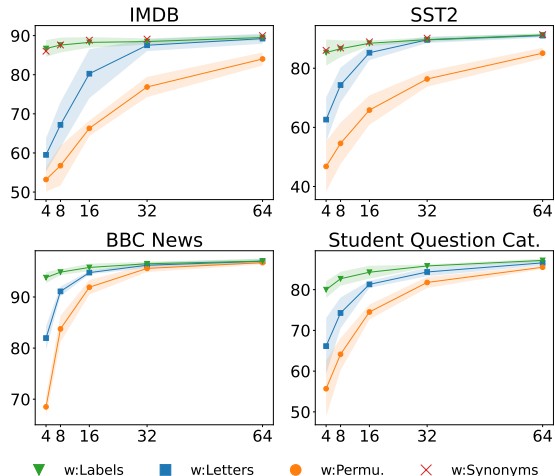

Figure 2: Ablation study: average accuracy, $\pm$ std deviation for different training sizes and anchor statements. 'w:synonyms' lines omitted to avoid overplotting.

is better than providing it with no information (letters), 2) confusing the model with opposite labels is worse than no information. Both indicate that the model indeed benefits from semantic information provided in the anchor statements. Especially, we observe large numeric differences in the accuracy score for training set sizes of 4 and 8. Thus, in the more extreme few-shot settings, the anchor statements in AncSetFit provide the greatest benefit. The larger the training set size, the less the model depends on correct semantic information in the anchor statement, as the performance gaps close. Intuitively this makes sense, as the more training data the easier to infer the commonalities of the classes. We also observe, in the two tasks of sentiment analysis, that providing synonym words does not have a large effect, indicating that the method is robust to small semantic changes in the template.

# 7 Conclusion

This paper proposes AncSetFit, an efficient method for extreme few-shot text classifications on small model sizes. The method injects semantic information about the task and labels to fine-tune a Sentence Transformer. The method separates samples of different classes in the embedding space by providing anchor points of textual information about the class labels to cluster around. We show in a number of tasks for sentiment and topic classifications that AncSetFit achieves higher performance in the extreme few-shot settings where there are fewer than eight training samples per class.

## Limitations

The method is limited to text classification tasks, where the task and corresponding classes can somehow be described with short opposing textual statements. Therefore, when tasks become more semantically complex, i.e., not easily described in a sentence, then the method approaches its limits. Additionally, the method's advantage is the relatively high performance using small backbone model sizes, but of course, like SetFit it still has a performance gap to large-scale models (Tunstall et al., 2022).

## Ethics Statement

As ever, generalisations drawn from fewer points of data have a higher risk of being inaccurate. While methods like AncSetFit that improve quantitative performance on benchmarks based on fewer data points represent advances in data efficiency, it is important not to over-interpret higher scores as an indication of excellent broad-domain generalisation.

## Acknowledgements

This work was supported by the Danish Data Science Academy, which is funded by the Novo Nordisk Foundation (NNF21SA0069429) and VIL-LUM FONDEN (40516).

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

# A    Appendix: Visualisation of the method

We illustrate the AncSetFit and the effect of its anchoring approach through PCA on the Emotion and CR datasets. In each task, we illustrate the two principal components of the embeddings on 1000 test samples before fine-tuning the Sentence Transformer model, and after fine-tuning with the AnchSetFit approach on only 4 instances per class. We show the anchor statements in the plot both before and after fine-tuning. On both the Emotion and the CR datasets, we can observe that the samples and anchor statements of the different classes separate more in the space after fine-tuning. Visually, we can imagine how the anchor statement is pushed apart and how the samples are drawn towards the anchor statement. Illustrating how we expect the method to work.

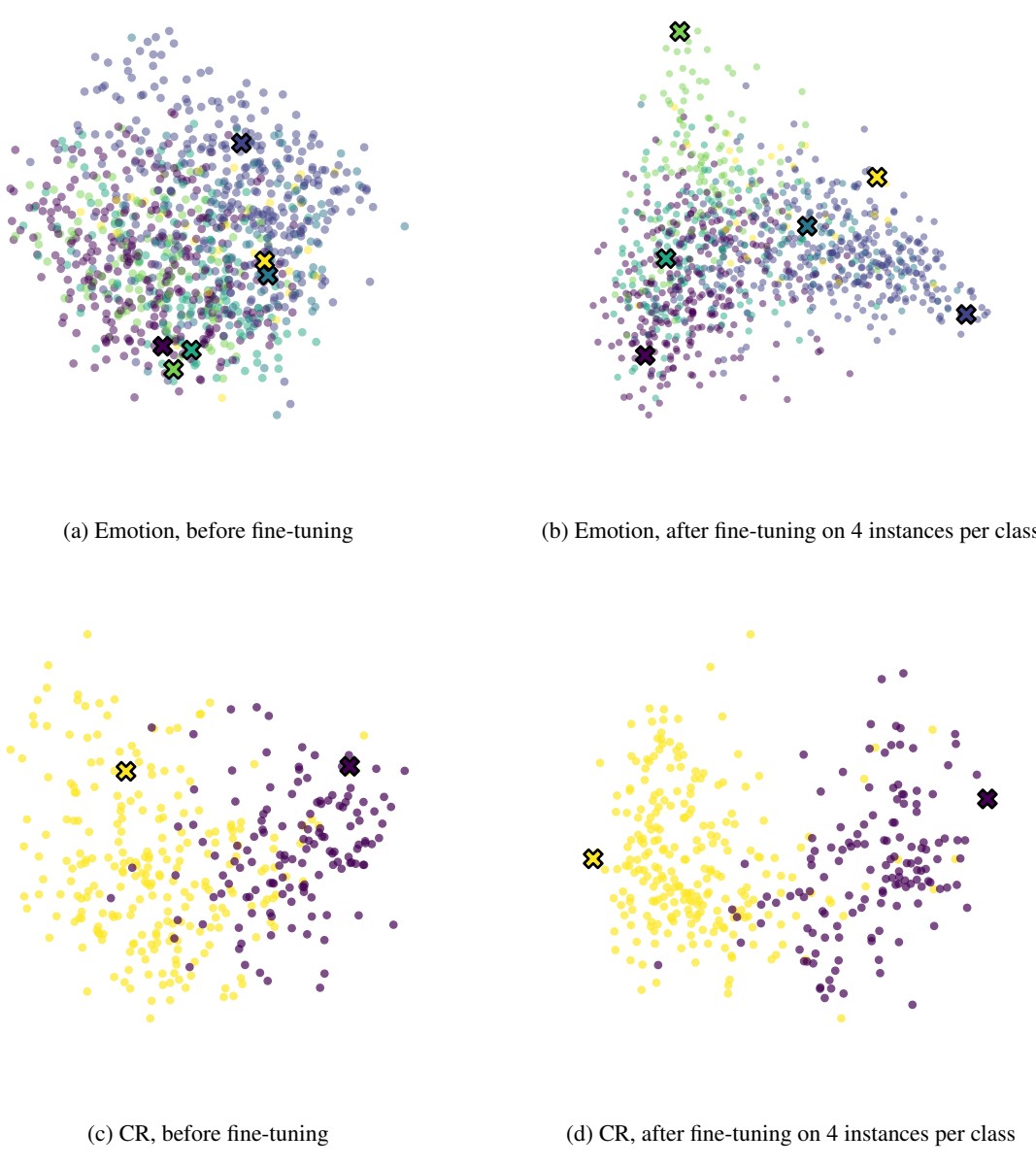

(a) Emotion, before fine-tuning

(b) Emotion, after fine-tuning on 4 instances per class

(c) CR, before fine-tuning

(d) CR, after fine-tuning on 4 instances per class

Figure 3: Illustration using PCA on embeddings of the AncSetFit mehtod

# B    Appendix: Datasets

Table 3 lists the datasets used respectively in the development phase and for testing. We provide a short task overview, the number of classes and the support of the testing set, the *) means we

have limited the test-set size. The majority of datasets are downloaded from `https://huggingface.co/SetFit`, where we follow any split or partition of the data provided here. Other sources of data are 1) Yahoo Questions downloaded at `https://huggingface.co/datasets/yahoo_answers_topics`, 2) Abstract which is from a Kaggle competition at `https://kaggle.com/datasets/vin1234/janatahack-independence-day-2020-ml-hackathon`, and here the test set is taken out of the train part for the experiments. 3) the english part SemEval2018Task1 is origional from Mohammad et al. (2018) and downloaded from `https://huggingface.co/datasets/sem_eval_2018_task_1`. Notice, the majority of the datasets are single-label tasks, except GoEmotions (Demszky et al., 2020), Abstract and semeval2018task1 which are multi-label tasks.

|  |  | Task | #Classes | Test size |
|---|---|---|---|---|
| **Development** | SST2 | Sentiment, movie review | 2 | 1821 |
|  | IMDB | Sentiment, movie review | 2 | 25000 |
|  | Student Question Catagories | Topic, subject of question | 4 | 5000 |
|  | BBC News | Topic, news | 5 | 1000 |
|  | TREC-QC | Topic, questions | 50 | 500 |
|  | GoEmotons | Multi-label sentiment, emotions on Reddit | 28 | 5427 |
|  | Abstract | Multi-label topic, research articles | 6 | 5243 |
| **Testing** | CR | Sentiment, customer reviews on electronics | 2 | 376 |
|  | SST5 | Sentiment, movie reviews | 5 | 2210 |
|  | Emotion | Sentiment, emotions in tweets | 6 | 2000 |
|  | AG News | Topic, news | 4 | 7600 |
|  | Yahoo Questions | Topic, questions | 10 | 5000* |
|  | Yelp Review | Sentiment, reviews | 5 | 5000* |
|  | Semeval2018task1 | Multi-label sentiment, affects in tweets (EN) | 11 | 3259 |

Table 3

## C Appendix: Anchor Statements

In the experiments, the anchor statement is generated following the generic form

$$template + \{w\} \tag{1}$$

where $w$ represent a describing word(s) of each particular class. In the multi-label cases, the anchor statements are generated per train sample by including all relevant class' descriptions

$$template + w_i, ..., w_n \tag{2}$$

for $i = 1, 2, .., n$ where $n$ is number of clases in the sample.
The listing below shows the template and class words for the different test tasks.

```
TEMPLATE = {
    SST5: "The movie is {}",
    CR: "The movie is {}",
    Emotion: "The emotion is {}",
    AG News: "The topic is about {}",
    Yelp Review: "The experience was {}",
    Yahoo Questions: "The question is about {}",
    SemEval2018Task1: "The emotions are {}",
}
```

```
words= {
    SST5: ["terrible", "bad", "okay", "good", "great"],
    Emotion: ["sadness","joy","love","anger","fear","surprise"],
    CR: ["bad","good"],
    AG News: ["World", "Sports", "Business", "Sci/Tech"],
    Yelp Review: ["terrible", "bad", "okay", "good", "great"],
    Yahoo Questions: ["Society & Culture",
    "Science & Mathematics", "Health", "Education & Reference",
    "Computers & Internet", "Sports", "Business & Finance",
    "Entertainment & Music", "Family & Relationships",
    "Politics & Government"],
    SemEval2018Task1: ['anger', 'anticipation', 'disgust', 'fear',
    'joy', 'love', 'optimism', 'pessimism', 'sadness', 'surprise',
    'trust']
}
```

Table 4 shows the anchor statement used in the ablation study described in Section 6 and corresponds to the results shown in Figure 2.

| | Template | $w$:labels | $w$: letters | $w$: synonyms |
|---|---|---|---|---|
| SST2 | "The movie review is " | ['negative','positive'] | ['A', 'B'] | ['bad','good'] |
| IMDB | "The movie review is " | ['negative','positive'] | ['A', 'B'] | ['bad','good'] |
| BBC News | "The topic is about " | ['tech', 'business', 'sport', 'entertainment', 'politics'] | ['A', 'B', 'C', 'D','E'] | |
| SQC | "The subject is " | ['Biology', 'Chemistry', 'Maths', 'Physics'] | ['A', 'B', 'C', 'D'] | |

Table 4: Template and words to generate anchor statement for different tasks in the ablation study. Note, the $w$: Permutation of labels is a permutation of the labels shown under $w$:labels such that the class is assigned a wrong text label.

## D Appendix: Implementation and Training details

**SetFit**   We used the script implementation from SetFit code base[3], and follow the settings in Tunstall et al. (2022). We experiment with the Sentence Transformer (Reimers and Gurevych, 2019) backbone model `paraphrase-mpnet-base-v2` available at HuggingFace (Wolf et al., 2019) [4]. We generate pairs for fine-tuning by iterating 20 times over each train instance and pairing each with a sample of the same class and one of a different class. This creates a total of instances*2*20 pairs for fine-tuning. We fine-tune for 1 epoch, batch size of 16, and max-sequence length of 256, and use logistic regression as the classification head as in Tunstall et al. (2022).

**AncSetFit**   AncSetFit is implemented based on SetFit implementation. For easier comparison, we keep the hyperparameters as close as possible (see above). When generating train pairs, we iterate 40 times over the train instances, and for each instance, we construct the anchor statement for the particular class and sample an instance from a different class. We set the iteration parameters to 40, such the total number of train pairs for fine-tuning becomes instances*40 pairs, as in SetFit. The loss function's margin parameter $\epsilon = 0.25$ for single-label tasks and $\epsilon = 0.5$ for multi-label tasks, based on the development phase. The experiments are run with the anchor statements provided in Appendix C.

**ADAPET**   We use the ADAPET method as modified in Tunstall et al. (2022) and implemented in the SetFit code base[5]. The modification is that in these experiments, there is no access to a development set,

---

[3] https://github.com/huggingface/setfit

[4] https://huggingface.co/sentence-transformers/paraphrase-mpnet-base-v2

[5] https://github.com/huggingface/setfit

and therefore no way of choosing the best checkpoint, which is therefore disabled. The training runs for 1000 batches. Experiments are conducted both with the backbone model `albert-xxlarge-v2` [6] and with the smaller model `albert-base-v2` [7]. For verbaliser and patterns, we use: "[TEXT] [template] [words]", where templates and words for the different tasks can be accessed in Appendix C.

**PERFECT**    We run PERFECT on the six different datasets using the codebase from SetFit [8] fork from PERFECT codebase [9] with the standard hyperparameters and backbone model Roberta-large [10] (Liu et al., 2019) - we modify the code to add the datasets from Yahoo Questions and Yelp Review.

## E    Appendix: Running times

We compare running times, combining both training and inference times, for AncSetFit and ADAPET when training on 8 samples per class, running the script for one seed.

|         | Time ADAPET | Time AncSetFit |
|---------|-------------|----------------|
| **CR**      | 134 min | 14 sec |
| **SST5**    | 150 min | 22 sec |
| **Emotion** | 136 min | 23 sec |
| **AG News** | 202 min | 31 sec |

The characteristics of the machine used are,

```
Intel Core i9 10940X 3.3GHz 14-Core
MSI GeForce RTX 3090  2 STK
2 x 128GB RAM,
```

running Ubuntu 20.04.4 LTS.

---

[6]https://huggingface.co/albert-xxlarge-v2
[7]https://huggingface.co/albert-base-v2
[8]https://github.com/SetFit/perfect/
[9]https://github.com/facebookresearch/perfect
[10]https://huggingface.co/roberta-large