# OpenReview forum: "Anchoring Fine-tuning of Sentence Transformer with Semantic Label Information for Efficient Truly Few-shot Classification"
_EMNLP/2023/Conference — EMNLP 2023 Main_

### Official Review · Reviewer_Dzxn · 2023-07-31

**Soundness:** 3

**Excitement:**

2: Mediocre: This paper makes marginal contributions (vs non-contemporaneous work), so I would rather not see it in the conference.

**Paper Topic And Main Contributions:**

This paper addressed semantic information of SetFit, which aims to improve the performance in extremely few-shot scenario.

**Reasons To Accept:**

This work utilizes semantic label Information and improves the performance of SetFit.

**Reasons To Reject:**

This work aims to improve SetFit but compared with experimental design in SetFit, existing results in this work fail to demonstrate AncSetFit effectiveness.

**Reproducibility:**

5: Could easily reproduce the results.

**Reviewer Confidence:**

3: Pretty sure, but there's a chance I missed something. Although I have a good feel for this area in general, I did not carefully check the paper's details, e.g., the math, experimental design, or novelty.

---

> ### Author Rebuttal · Authors · 2023-08-29
>
> We like to thank the reviewer for taking the time to review our work.
>
> Our work aims at improving few-shot text classification in the cases where we both have extremely few data instances (2-8) and small model sizes, making the method resource light to run. Hence AncSetFit and SetFit can even be trained on a personal PC. This is why, we have focused mainly in the paper on comparing to SetFit (lines 231-235), as other methods use larger backbone models and more compute resources (see lines 270-274). ADAPET is getting better results in terms of accuracy in some of the tasks, but it also uses a backbone model with twice the number of parameters in the experiments (albert-xxlarge-v2), and a large factor more running time (see appendix, line 510-517).
>
>  We think we could make this point clearer in the paper by including more experiments where we compare to ADAPET with a similar backbone model size in terms of parameters as AncSetFit is using.
>  We also propose to include more comparisons to another few-shot method. We propose to include the following,
>
>  1) Results on ADAPET using ‘albert-base-v2’, together with a table comparing the different model’s parameter sizes.
>  2) Results with the few-shot method PERFECT (Rabeeh et al. 2022)
>  3) Results with ADAPET (albert-xxlarge-v2) on the two datasets Yahoo Questions and Yelp Review (ADAPET_xxlarge)
>
> Time did not allow us to complete all the experiments, but below you can see the results we obtained so far, note we only had time to run 5 seeds for all the newly added experiments, but plan to run 10. We observe,
>
> -   AncSetFit is outperforming PERFECT in all the tasks completed.
> -   AncSetFit is outperforming ADAPET_base in all the tasks completed, showing that with an equal small model size, AncSetFit is superior
> -   We only had time to run ADAPET_xxlarge for size 2 for Yahoo Answers and Yelp review, and here AncSetFit is getting higher accuracy on Yahoo, but lower than ADAPET_xxlarge for Yelp.
>
> | **Method**  | **Backbone**             | **Size** |
> |--------------------|---------------------------------|-----------------|
> |     PERFECT        |     roberta-large               |     355M        |
> |     ADAPET_xxl     |     albert-xxlarge-v2           |     223M        |
> |     ADAPET_base    |     albert-base-v2              |     117M        |
> |     SetFit         |     paraphrase-mpnet-base-v2    |     110M        |
> |     AncSetfit      |     paraphrase-mpnet-base-v2    |     110M        |
>
>
>
>  **Result table**
>
> |             | Method/size    | 2    | 4    | 8    |
> |-------------|----------------|------|------|------|
> | **Emotion**     | PERFECT        | 20.5 | 19.6 | 27.5 |
> |             | ADAPET_xxlarge | 46.7 | 48.1 | 53.9 |
> |             | ADAPET_base    | 37.5 | 43.6 | 49.3 |
> |             | SetFit         | 30.0 | 39.1 | 49.9 |
> |             | AncSetFit      | 49.1 | 53.2 | 56.7 |
> | -|
> | **AG News**     | PERFECT        | 49.5 | 69.6 | 80.0 |
> |             | ADAPET_xxlarge | 76.8 | 84.1 | 86.0 |
> |             | ADAPET_base    | 64.4 | 72.8 | 78.0 |
> |             | SetFit         | 65.3 | 76.5 | 82.8 |
> |             | AncSetFit      | 76.0 | 81.6 | 84.2 |
> | -|
> | **SST-5**      | PERFECT        | 25.8 | 32.0 | 36.2 |
> |             | ADAPET_xxlarge | 47.0 | 48.1 | 49.7 |
> |             | ADAPET_base    | 32.2 | 35.3 | 40.2 |
> |             | SetFit         | 33.4 | 38.4 | 42.3 |
> |             | AncSetFit      | 40.5 | 43.4 | 45.2 |
> | -|
> | **CR**          | PERFECT        | 62.2 | 76.6 | 81.1 |
> |             | ADAPET_xxlarge | 87.2 | 90.6 | 90.6 |
> |             | ADAPET_base    | 78.5 | 77.1 | 78.7 |
> |             | SetFit         | 75.5 | 86.8 | 89.6 |
> |             | AncSetFit      | 86.6 | 89.8 | 90.7 |
> |- |
> | **Yahoo Answers**       | ADAPET_xxlarge | 52.0 | p    | p    |
> |             | ADAPET_base    | 46.8 | 57.1 | 59.3 |
> |             | SetFit         | 42.9 | 55.4 | 61.0 |
> |             | AncSetFit      | 57.1 | 62.2 | 65.3 |
> | -|
> | **Yelp review** | ADAPET_xxlarge | 52.1 | p    | p    |
> |             | ADAPET_base    | 32.2 | 45.2 | p |
> |             | SetFit         | 36.6 | 44.9 | 50.7 |
> |             | AncSetFit      | 48.1 | 51.6 | 53.9 |
>
> *p: pending
>
> **References**
>
> Rabeeh Karimi Mahabadi, Luke Zettlemoyer, James Henderson, Lambert Mathias, Marzieh Saeidi, Veselin Stoyanov, and Majid Yazdani. 2022. Prompt-free and efficient few-shot learning with lan-
> guage models. In Proceedings of the 60th Annual Meeting of the Association for Computational Lin-
> guistics (Volume 1: Long Papers), pages 3638–3652,Dublin, Ireland. Association for Computational Linguistics.

---

### Official Review · Reviewer_qm4T · 2023-08-01

**Soundness:** 4

**Excitement:**

3: Ambivalent: It has merits (e.g., it reports state-of-the-art results, the idea is nice), but there are key weaknesses (e.g., it describes incremental work), and it can significantly benefit from another round of revision. However, I won't object to accepting it if my co-reviewers champion it.

**Paper Topic And Main Contributions:**

What this paper about?
- This paper tackles Few-Shot Text Classification.

What problem or question did this paper address?
- The first research question (RQ 1) is if the efficiency of small models in the SetFit method can be retained while also making the method label-aware and if this improves performance, especially in cases with little training data (Line 31-35).
- The second research question (RQ 2) is to what extent variation in textual information affects model performance (Line 40-41).

What are the main contributions that this paper makes towards a solution or answer?
- For RQ 1: In order to inform the model of what is to be classified, the authors proposes to add semantic information about the task and labels.
- For RQ 2: The authors used four variations of constructing anchor statement and demonstrated the impacts of the variations on the performances (Figure 1).


**Reasons To Accept:**

- This paper proposes a simple but effective few-shot text classification method.

**Reasons To Reject:**

- Compared with ADAPET, the performance of their proposed method is lower or on par, although their method has some merits over ADAPET.

**Reproducibility:**

4: Could mostly reproduce the results, but there may be some variation because of sample variance or minor variations in their interpretation of the protocol or method.

**Reviewer Confidence:**

3: Pretty sure, but there's a chance I missed something. Although I have a good feel for this area in general, I did not carefully check the paper's details, e.g., the math, experimental design, or novelty.

---

> ### Author Rebuttal · Authors · 2023-08-29
>
> We like to thank the reviewer for taking the time to review our work.
>
> Our work aims at improving few-shot text classification in the cases where we both have extremely few data instances (2-8) and small model sizes, making the method resource light to run. Hence AncSetFit and SetFit can even be trained on a personal PC. This is why, we have focused mainly in the paper on comparing to SetFit (lines 231-235), as other methods use larger backbone models and more compute resources (see lines 270-274). ADAPET is getting better results in terms of accuracy in some of the tasks, but it also uses a backbone model with twice the number of parameters in the experiments (albert-xxlarge-v2), and a large factor more running time (see appendix, line 510-517).
>
>  We think we could make this point clearer in the paper by including more experiments where we compare to ADAPET with a similar backbone model size in terms of parameters as AncSetFit is using.
>  We also propose to include more comparisons to another few-shot method. We propose to include the following,
>
>  1) Results on ADAPET using ‘albert-base-v2’, together with a table comparing the different model’s parameter sizes.
>  2) Results with the few-shot method PERFECT (Rabeeh et al. 2022)
>  3)Results with ADAPET (albert-xxlarge-v2) on the two datasets Yahoo Questions and Yelp Review (ADAPET_xxlarge)
>
> Time did not allow us to complete all the experiments, but below you can see the results we obtained so far, note we only had time to run 5 seeds for all the newly added experiments, but plan to run 10. We observe,
>
> -   AncSetFit is outperforming PERFECT in all the tasks completed.
> -   AncSetFit is outperforming ADAPET_base in all the tasks completed, showing that with an equal small model size, AncSetFit is superior
> -   We only had time to run ADAPET_xxlarge for size 2 for Yahoo Answers and Yelp review, and here AncSetFit is getting higher accuracy on Yahoo, but lower than ADAPET_xxlarge for Yelp.
>
> | **Method**  | **Backbone**             | **Size** |
> |--------------------|---------------------------------|-----------------|
> |     PERFECT        |     roberta-large               |     355M        |
> |     ADAPET_xxl     |     albert-xxlarge-v2           |     223M        |
> |     ADAPET_base    |     albert-base-v2              |     117M        |
> |     SetFit         |     paraphrase-mpnet-base-v2    |     110M        |
> |     AncSetfit      |     paraphrase-mpnet-base-v2    |     110M        |
>
>
>
>  **Result table**
>
> |             | Method/size    | 2    | 4    | 8    |
> |-------------|----------------|------|------|------|
> | **Emotion**     | PERFECT        | 20.5 | 19.6 | 27.5 |
> |             | ADAPET_xxlarge | 46.7 | 48.1 | 53.9 |
> |             | ADAPET_base    | 37.5 | 43.6 | 49.3 |
> |             | SetFit         | 30.0 | 39.1 | 49.9 |
> |             | AncSetFit      | 49.1 | 53.2 | 56.7 |
> | -|
> | **AG News**     | PERFECT        | 49.5 | 69.6 | 80.0 |
> |             | ADAPET_xxlarge | 76.8 | 84.1 | 86.0 |
> |             | ADAPET_base    | 64.4 | 72.8 | 78.0 |
> |             | SetFit         | 65.3 | 76.5 | 82.8 |
> |             | AncSetFit      | 76.0 | 81.6 | 84.2 |
> | -|
> | **SST-5**      | PERFECT        | 25.8 | 32.0 | 36.2 |
> |             | ADAPET_xxlarge | 47.0 | 48.1 | 49.7 |
> |             | ADAPET_base    | 32.2 | 35.3 | 40.2 |
> |             | SetFit         | 33.4 | 38.4 | 42.3 |
> |             | AncSetFit      | 40.5 | 43.4 | 45.2 |
> | -|
> | **CR**          | PERFECT        | 62.2 | 76.6 | 81.1 |
> |             | ADAPET_xxlarge | 87.2 | 90.6 | 90.6 |
> |             | ADAPET_base    | 78.5 | 77.1 | 78.7 |
> |             | SetFit         | 75.5 | 86.8 | 89.6 |
> |             | AncSetFit      | 86.6 | 89.8 | 90.7 |
> |- |
> | **Yahoo Answers**       | ADAPET_xxlarge | 52.0 | p    | p    |
> |             | ADAPET_base    | 46.8 | 57.1 | 59.3 |
> |             | SetFit         | 42.9 | 55.4 | 61.0 |
> |             | AncSetFit      | 57.1 | 62.2 | 65.3 |
> | -|
> | **Yelp review** | ADAPET_xxlarge | 52.1 | p    | p    |
> |             | ADAPET_base    | 32.2 | 45.2 | p |
> |             | SetFit         | 36.6 | 44.9 | 50.7 |
> |             | AncSetFit      | 48.1 | 51.6 | 53.9 |
>
> *p: pending
>
> **References**
>
> Rabeeh Karimi Mahabadi, Luke Zettlemoyer, James Henderson, Lambert Mathias, Marzieh Saeidi, Veselin Stoyanov, and Majid Yazdani. 2022. Prompt-free and efficient few-shot learning with lan-
> guage models. In Proceedings of the 60th Annual Meeting of the Association for Computational Lin-
> guistics (Volume 1: Long Papers), pages 3638–3652,Dublin, Ireland. Association for Computational Linguistics.

---

### Official Review · Reviewer_3p6y · 2023-08-02

**Typos Grammar Style And Presentation Improvements:** Consider moving the the ablation stud…
**Soundness:** 3

**Excitement:**

3: Ambivalent: It has merits (e.g., it reports state-of-the-art results, the idea is nice), but there are key weaknesses (e.g., it describes incremental work), and it can significantly benefit from another round of revision. However, I won't object to accepting it if my co-reviewers champion it.

**Paper Topic And Main Contributions:**

The paper focuses on the low-data few-shot scenario in text classification. The authors extend a previous work (SetFit) by introducing semantic information for the labels, specifically they generate triplets with an "anchor statement" to incorporate the information into the loss. The proposed method (AncSetFit) is then tested on text classification datasets.

**Questions For The Authors:**

A) Why is ADAPET only reported on the table and not on the graphs?

B) Why is the ablation data different from the data of the experimental results?

**Reasons To Accept:**

AncSetFit improves the original work (SetFit) without being more computationally demanding.

**Reasons To Reject:**

The experimental comparisons are lacking. The authors focuses on the comparison with SetFit but neglects most other methods. They only report ADAPET as comparison (and only for some of the datasets) which performs better than AncSetFit in 3 out of the 4 instances.
Moreover, the paper of SetFit has more comparisons that in AncSetFit have not been considered at all.

**Reproducibility:**

4: Could mostly reproduce the results, but there may be some variation because of sample variance or minor variations in their interpretation of the protocol or method.

**Reviewer Confidence:**

4: Quite sure. I tried to check the important points carefully. It's unlikely, though conceivable, that I missed something that should affect my ratings.

---

> ### Author Rebuttal · Authors · 2023-08-29
>
> We like to thank you for taking the time to review our work and for the specific comments by which we can improve our work.
> Our work aims at improving few-shot text classification in the cases where we both have extremely few data instances (2-8) and small model sizes, making the method resource light to run. Hence AncSetFit and SetFit can even be trained on a personal PC. This is why, we have focused mainly in the paper on comparing to SetFit (lines 231-235), as other methods use larger backbone models and more compute resources (see lines 270-274). ADAPET is getting better results in terms of accuracy in some of the tasks, but it also uses a backbone model with twice the number of parameters in the experiments (albert-xxlarge-v2), and a large factor more running time (see appendix, line 510-517).
>
> We agree with your point, that our work could benefit from more comparisons. We are therefore suggesting to include the following in the paper:
> 1) Results with the few-shot method PERFECT (Rabeeh et al. 2022)
> 2) Results where ADAPET is using a backbone model (albert-base-v2)  which is equal in terms of model parameters as the backbone model used in AncSetFit and Setfit (ADAPET_base)
> 3) Results with ADAPET (albert-xxlarge-v2) on the two  datasets Yahoo Questions and Yelp Review (ADAPET_xxlarge)
>
> We would then also include a table showing the different number of model parameters used in each methods, such it easier to compare the methods both in terms of accuracy and sizes (see below).  We did not have time to run all the experiments in the author response period, but we have included the finished one here, all the new experiments is ran for 5 seeds, but we plan to run 10 seeds, we can see the following:
>
> -	AncSetFit is outperforming PERFECT in all the tasks completed.
> -	AncSetFit is outperforming ADAPET_base in all the tasks completed, showing that with an equal small model size, AncSetFit is superior
> -	We only had time to run ADAPET_xxlarge for size 2 for Yahoo Answers and Yelp review, and here AncSetFit is higher accurcay on Yahoo, but lower then ADAPET_xxlarge for Yelp.
>
>
> | **Method**  | **Backbone**             | **Size** |
> |--------------------|---------------------------------|-----------------|
> |     PERFECT        |     roberta-large               |     355M        |
> |     ADAPET_xxl     |     albert-xxlarge-v2           |     223M        |
> |     ADAPET_base    |     albert-base-v2              |     117M        |
> |     SetFit         |     paraphrase-mpnet-base-v2    |     110M        |
> |     AncSetfit      |     paraphrase-mpnet-base-v2    |     110M        |
> <br />
>
>  **Result table**
>
> |             | Method/size    | 2    | 4    | 8    |
> |-------------|----------------|------|------|------|
> | **Emotion**     | PERFECT        | 20.5 | 19.6 | 27.5 |
> |             | ADAPET_xxlarge | 46.7 | 48.1 | 53.9 |
> |             | ADAPET_base    | 37.5 | 43.6 | 49.3 |
> |             | SetFit         | 30.0 | 39.1 | 49.9 |
> |             | AncSetFit      | 49.1 | 53.2 | 56.7 |
> | -|
> | **AG News**     | PERFECT        | 49.5 | 69.6 | 80.0 |
> |             | ADAPET_xxlarge | 76.8 | 84.1 | 86.0 |
> |             | ADAPET_base    | 64.4 | 72.8 | 78.0 |
> |             | SetFit         | 65.3 | 76.5 | 82.8 |
> |             | AncSetFit      | 76.0 | 81.6 | 84.2 |
> | -|
> | **SST-5**      | PERFECT        | 25.8 | 32.0 | 36.2 |
> |             | ADAPET_xxlarge | 47.0 | 48.1 | 49.7 |
> |             | ADAPET_base    | 32.2 | 35.3 | 40.2 |
> |             | SetFit         | 33.4 | 38.4 | 42.3 |
> |             | AncSetFit      | 40.5 | 43.4 | 45.2 |
> | -|
> | **CR**          | PERFECT        | 62.2 | 76.6 | 81.1 |
> |             | ADAPET_xxlarge | 87.2 | 90.6 | 90.6 |
> |             | ADAPET_base    | 78.5 | 77.1 | 78.7 |
> |             | SetFit         | 75.5 | 86.8 | 89.6 |
> |             | AncSetFit      | 86.6 | 89.8 | 90.7 |
> |- |
> | **Yahoo Answers**       | ADAPET_xxlarge | 52.0 | p    | p    |
> |             | ADAPET_base    | 46.8 | 57.1 | 59.3 |
> |             | SetFit         | 42.9 | 55.4 | 61.0 |
> |             | AncSetFit      | 57.1 | 62.2 | 65.3 |
> | -|
> | **Yelp review** | ADAPET_xxlarge | 52.1 | p    | p    |
> |             | ADAPET_base    | 32.2 | 45.2 | p |
> |             | SetFit         | 36.6 | 44.9 | 50.7 |
> |             | AncSetFit      | 48.1 | 51.6 | 53.9 |
>
> *p: pending
>
> **Question A)**  On the graphs we compare SetFit and AncSetFit for training data sizes between 2-64. With the graphs, we show, that AncSetFit only has an advantage when we a dealing with really small training sizes (2-8). Our comparison with ADAPET therefore only focuses on data sizes between 2-8. We therefore did not plot it as we do not have results for the whole range. Comparing  ADAPET with SetFit for training data size of 64 can be seen in Tunstall et al. (2022).
>
> **Question B)** We wanted to follow a setup like in Tunstall et al. (2022), where we start by splitting a list of datasets into either development datasets or testing datasets. The idea is to do no hyperparameter-tuning on the test dataset in order to mimic a real few-shot settings as best as possible (See 167-170, 185-189 ). In the ablation study, we changed the templates and examined the effect of this. We did it on the development datasets, because we wanted to only have “one go” on the test datasets. We therefore also ran these experiments before we did any testing.
>
> We do see the point, that this is not very clear in the text. We think it is a good idea to move the section to after the results and make it more clear.
>
> **References**
>
> Rabeeh Karimi Mahabadi, Luke Zettlemoyer, James Henderson, Lambert Mathias, Marzieh Saeidi, Veselin Stoyanov, and Majid Yazdani. 2022. Prompt-free and efficient few-shot learning with lan-
> guage models. In Proceedings of the 60th Annual Meeting of the Association for Computational Lin-
> guistics (Volume 1: Long Papers), pages 3638–3652,Dublin, Ireland. Association for Computational Linguistics.
>
> Lewis Tunstall, Nils Reimers, Unso Eun Seo Jo, Luke
> Bates, Daniel Korat, Moshe Wasserblat, and Oren
> Pereg. 2022. Efficient few-shot learning without
> prompts. arXiv preprint arXiv:2209.11055.

---

### Meta-Review · Area_Chair_aE5C · 2023-09-18

**Recommendation:** 3

**Metareview:**

The paper introduces AncSetFit, a method for improving few-shot text classification in resource-constrained scenarios. It incorporates semantic label information and focuses on scenarios with minimal training data and small model sizes, aiming for efficiency. The authors conducted experiments on various datasets, comparing AncSetFit to SetFit and ADAPET. Reviewers appreciated the simplicity and efficiency of the proposed method but raised concerns about limited comparisons and the performance relative to ADAPET. While the paper's contributions are smaller due to it being a short-paper, the authors have taken steps to address the reviewers' concerns and enhance the comprehensiveness of their work.

**Key Concerns of Reviewers:**

Reviewers expressed concerns about the paper's limited experimental scope, focusing mainly on comparisons with SetFit and not including comparisons with other methods.
There were concerns about the performance comparison with ADAPET, which outperformed AncSetFit in some instances, possibly due to differences in model size and computational resources.
The authors responded by proposing additional experiments and comparisons, including results with ADAPET using a smaller backbone model and introducing results with the PERFECT method.

The authors are encouraged to extend the abstract with 1-2 sentences that briefly describe the proposed method in more detail.

**Note:** The review and scores of reviewer "@Dzxn" need to be ignored due to the low-quality review.

---

### Decision · Program_Chairs · 2023-10-07

**Decision:**

Accept-Main

**Comment:**

The paper introduces AncSetFit, a method for improving few-shot text classification in resource-constrained scenarios. It incorporates semantic label information and focuses on scenarios with minimal training data and small model sizes, aiming for efficiency. The authors conducted experiments on various datasets, comparing AncSetFit to SetFit and ADAPET. Reviewers appreciated the simplicity and efficiency of the proposed method but raised concerns about limited comparisons and the performance relative to ADAPET. While the paper's contributions are smaller due to it being a short-paper, the authors have taken steps to address the reviewers' concerns and enhance the comprehensiveness of their work.

**Key Concerns of Reviewers:**

Reviewers expressed concerns about the paper's limited experimental scope, focusing mainly on comparisons with SetFit and not including comparisons with other methods.
There were concerns about the performance comparison with ADAPET, which outperformed AncSetFit in some instances, possibly due to differences in model size and computational resources.
The authors responded by proposing additional experiments and comparisons, including results with ADAPET using a smaller backbone model and introducing results with the PERFECT method.

The authors are encouraged to extend the abstract with 1-2 sentences that briefly describe the proposed method in more detail.

**Note:** The review and scores of reviewer "@Dzxn" need to be ignored due to the low-quality review.